# Melanoma Immunotherapy and Precision Medicine in the Era of Tumor Micro-Tissue Engineering: Where Are We Now and Where Are We Going?

**DOI:** 10.3390/cancers13225788

**Published:** 2021-11-18

**Authors:** Francesca Varrone, Luigi Mandrich, Emilia Caputo

**Affiliations:** 1IRBM S.p.A., Via Pontina Km 30,600, I-00071 Pomezia, Italy; fvarrone80@gmail.com; 2Research Institute on Terrestrial Ecosystem—IRET-CNR Via Pietro Castellino 111, I-80131 Naples, Italy; luigi.mandrich@cnr.it; 3Institute of Genetics and Biophysics—IGB-CNR, “A. Buzzati-Traverso”, Via Pietro Castellino 111, I-80131 Naples, Italy

**Keywords:** melanoma, tumor microenvironment, multicellular spheroids, organotypic melanoma models, skin-on-chip

## Abstract

**Simple Summary:**

Melanoma is a cancer with very poor survival rates, although its treatment has been revolutionized by targeted therapy and immunotherapy. It is a complex disease, where dynamic interactions, not only among melanoma cells themselves, but also between tumor cells and their surroundings, occur. This review focuses on melanoma complexity, pointing out how melanoma cells are only a part of a large ecosystem and the tumor microenvironment plays an active role on their evolution and ability to escape to drug treatment. The recent efforts addressed to the development of ex-vivo micro-tissue models able to recapitulate the live conditions of melanoma cells in human patients have been outlined. Further, the use of ex-vivo models as a novel approach for the researcher to investigate the mechanisms underlying tumor biology and immunotherapeutic resistance in metastatic melanoma has been discussed, as well as their high potential for the development of personalized medicine in melanoma treatment.

**Abstract:**

Malignant melanoma still remains a cancer with very poor survival rates, although it is at the forefront of personalized medicine. Most patients show partial responses and disease progressed due to adaptative resistance mechanisms, preventing long-lasting clinical benefits to the current treatments. The response to therapies can be shaped by not only taking into account cancer cell heterogeneity and plasticity, but also by its structural context as well as the cellular component of the tumor microenvironment (TME). Here, we review the recent development in the field of immunotherapy and target-based therapy and how, in the era of tumor micro-tissue engineering, ex-vivo assays could help to enhance our melanoma biology knowledge in its complexity, translating it in the development of successful therapeutic strategies, as well as in the prediction of therapeutic benefits.

## 1. Introduction

### 1.1. Melanoma

Melanoma is the deadliest type of skin cancer. It represents about 5% of all skin tumors and it is the cause of more than 75% of skin cancer deaths worldwide. Patients carrying localized or regional disease show a 5-year relative survival rate value of 98% and 64%, respectively, while this value drastically decreases to 23% in metastatic patients [1].

Melanomas originate from the malignant transformation of the melanocytes and they are mainly classified in three subtypes according to the localization of melanocytes undergoing the transformation: 1. cutaneous melanoma (CM), from skin melanocytes; 2. uveal melanoma (UM), from melanocytes in the choroid, ciliary body, and iris of the eye, and 3. mucosal melanomas (MM) from melanocytes in mucosal membranes [2,3,4]. CM represents 91.2%, while UM 5.3% and MM 1.3% of all melanomas recorded in the USA. Due to its prevalence, CM (hereafter melanoma) is the most studied subtype among the three and it will be the focus of this review.

Melanoma is a complex disease consisting of a multistep process, involving the accumulation of genetic and/or epigenetic somatic modifications and exposition to environmental factors, where not only melanoma cells themselves, but constant interactions occurring between tumor cells and their surroundings play a crucial role in disease dissemination, therapy resistance, and mortality. Indeed, it is a dynamic process where inter- and intra-tumoral heterogeneity, phenotypic plasticity, stromal reprogramming, and microbiome [5] are among some of the key drivers of melanoma progression. Over the last decade, remarkable advances have been made in expounding melanoma etiology and pathogenesis, promoting the identification and validation of novel drug targets and biomarkers [6]. Today, surgery remains the gold standard treatment for the primary melanoma, mainly for thin melanomas, with low risk of dissemination and long-term survival benefits. On the other hand, metastatic melanoma therapy has been revolutionized by developments in targeted therapy and immunotherapy.

Targeted therapy approaches have been possible thanks to the identification of important somatic mutations, referred to as ‘driver mutations’, conferring advantages to melanoma growth and progression. The most studied and well-characterized driver mutations in melanoma have been found mainly in genes involved in the Mitogen-Activated Protein Kinase (MAPK) [7,8,9] and in the Phosphoinositide 3-Kinase/protein Kinase B (PI3K/AKT) pathways [10,11]. Among these genes, B-RAF, encoding the proto-oncogene serine/threonine kinase, has been the first well-studied driver mutation to be used in targeted therapy approach for melanoma treatment and, over time, different drugs have been developed against B-RAF mutated proteins, from vemurafenib to dabrafenib and the recent encorafenib. The most significant somatic driver mutations and the corresponding designed targeting drugs are reported in Table 1.

Immunotherapy is another efficient treatment option for metastatic melanoma patients, because of the high immunogenicity of this tumor. Four main groups of immunotherapeutic treatments are currently available for melanoma treatment [16], as illustrated in Figure 1.

In particular, antibodies directed to specific immune checkpoints such as anti-programmed cell death 1 (PD-1) and anti-cytotoxic T-lymphocyte-associated protein 4 (CTLA-4) brought a statistically significant benefit in terms of overall survival (OS), progression-free survival (PFS) and overall response rate (ORR) compared to chemotherapy.

Both target based therapeutics and the immune checkpoint inhibitors [23] have drastically changed the clinical management of melanoma and improved melanoma patient outcome, since their FDA approvals. The current immunotherapeutic drugs for melanoma treatment are summarized in Table 2.

However, they left a big challenge for investigators and clinicians in overcoming the drug resistance and disease recurrence issues responsible of the therapeutic plateau we are in today. 

Recent advances in tumor micro-tissue engineering are providing novel insights in the melanoma biology and in its complexity, which can be translated in the development of innovative and successful target-based and immune therapies.

### 1.2. Heterogeneity and Plasticity: The Most Striking Melanoma Properties

Melanoma is characterized by a high heterogeneity [24] and plasticity [25]. The intra-tumor heterogeneity is the product of the high irreversible genetic instability of melanoma cells as well as of their ability to undergo reversible phenotypic changes. The high genetic instability [8] generates the necessary genetic modification leading to the irreversible cell-intrinsic phenotype switching ability of melanoma cells, while on the other hand, their reversible ability to switch phenotype [26] is driven by micro-environmental cues. Both, cell-intrinsic and microenvironmental-driven, are more generally referred to as ‘phenotype switching’, which is, actually, the better model explaining the dynamic melanoma evolution [26], as explained below. It includes both the clonal evolution [27] and the cancer-stem cells [28] models.

Further, melanoma cells display an extreme plasticity. They are able to activate a plastic network of signal transduction pathways passing from one path to another one, in order to keep the continuous transmission of survival signals even in hostile environments. In addition, their ability to transdifferentiate to a variety of states under different circumstances represents a further mechanism underlying their incredible plasticity, as explained in the following sections.

Nevertheless, melanoma cells are able to influence the tumor microenvironment by a stromal reprogramming mechanism, which is responsible for their long-term growth and drug resistance.

Melanoma evolution and phenotype switching. Melanoma cells derive from the melanocytes transformation, which is accompanied by the accumulation of a high number of driver mutations, as demonstrated by cancer genome deep sequencing [29]. This genetic instability [8] shapes the necessary genetic alteration driving to the irreversible cell-intrinsic phenotype switching ability of melanoma cells responsible for melanoma progression.

Moreover, it has been demonstrated that melanoma progression is associated with the reactivation of melanocyte differentiation program [30]. Briefly, melanocytes are the product of an active epithelial to mesenchymal transition (EMT) program occurring during embryogenesis, in the (neuroepithelial) neural crest stem cells (NCSCs), despite the fact that melanocytes are not true epithelial cells. The NCSCs are a multipotent, migratory, and transient cell population. During embryonic development, these cells migrate through the vertebrate embryo and infiltrate different organs where they are capable to differentiate in various cell lineages including melanocytes [26]. It has been demonstrated that ZEB2, one among the EMT-inducing transcription factors (EMT-TFs), is essential for terminal differentiation in vivo of NCSCs in melanocytes, Schwann cells and oligodendrocytes, through the upregulation of microphthalmia-associated transcription factor, MITF [31,32].

EMT is a cellular program crucial not only during embryogenesis, but also in the course of fibrosis, wound healing and carcinogenesis [26,33]. It induces a down-regulation of epithelial markers and an upregulation of mesenchymal markers in the cells, accompanied by a morphological change from an epithelioid towards a mesenchymal/spindle cell shape as well as by a remodeling of cell-cell and cell-matrix interactions with subsequent enhanced cell motility and migration [34].

The molecular pathways activated by EMT-TFs and MITF, during the melanocyte differentiation program, may be reactivated during melanomagenesis, which explains melanoma heterogeneity and plasticity. It has been found that likely in NC-derived melanoblasts, the switch from E to N-Cadherin does occur in a subset of melanomas and it is induced by ZEB1, TWIST and SNAIL EMT-TFs [26,35,36]. These findings support that melanoma progression is not founded only on irreversible clonal or lineage-driven remodeling, but can be induced by reversible and functional reprograming of signaling pathways, activated by EMT-TFs and MITF, according to the ‘phenotype switching’ model as showed in Figure 1.

Moreover, it has been observed that CD271 and SOX10, two factors [37,38] associated with NCSC regulatory networks [30], are highly expressed in human melanoma and their expression is correlated with a high metastatic potential and a worse patient prognosis [39].

Despite the fact that melanoma phenotypic diversity and plasticity have been known for many years, the molecular characterization of the different phenotypic states has been pointed out with the cloning of MITF genes and with the discovery of its role in the reversible phenotype switching of melanoma cells between an MITF-positive/drug sensitive ‘proliferative’ state and an MITF-low/drug resilient ‘invasive’ cell state. Moreover, it has been observed that MITF-depleted cells showed a more stem cell-like phenotype, an increased plasticity and a reduced proliferation, promoting tumor progression, while cells expressing high levels of MITF stimulated proliferation and differentiation [34], as illustrated in Figure 1.

Single-cell RNA sequencing has further contributed to the identification of the transcriptional programs underlying melanoma phenotypic heterogeneity [40]. These studies confirmed the key role of MITF for distinct phenotypic states, in particular, its correlation to a differentiation gene-expression program. On the other hand, these studies highlighted the opposite role of AXL, a receptor tyrosine kinase belonging to TAM family in promoting the invasive, dedifferentiated drug resistant phenotype [41,42,43,44].

Interestingly, it has been observed that melanomas classified as MITF^High^ contained a small amount of cells expressing AXL^High^/MITF^Low^ program. This small population increased upon treatment with BRAF inhibitors (BRAFi) as single agents or in combination with MEK inhibitors (MEKi), along with a distinct resistant population of MET-high cells. Moreover, the AXL^High^/MITF^Low^ cell population was associated with increased numbers of cancer associated fibroblasts (CAFs), while MITF^High^ melanomas showed a reduced CAF infiltration.

Actually, the phenotypic states of melanoma cells are not limited to an MITF-positive/drug sensitive ‘proliferative’ and an MITF-low/drug resilient ‘invasive’ cell state; however, to date, a revised MITF rheostat model, including six different phenotypic states, has been reported.

As shown in Figure 1, the states are ranked in relation to MITF and SOX10 expression. Starting from the MITF^Low^/SOX10^Low^ undifferentiated state, the most undifferentiated melanoma cells, having lost expression of both melanocytic transcription factors, SOX10 and MITF, to the MITF^Low^/SOX10^Medium^ neural crest stem cell (NCSC) state, to the MITF^Medium^/SOX10^Medium^ starved melanoma cell (SMC), intermediate and melanocytic states to the MITF^High^/SOX10^Low^ hyper-differentiated state. This classification, according to the MITF expression or differentiation, does not rule out the potential of each plastic state to generate another one. Indeed, it has been demonstrated that hyper-differentiated, NCSC and undifferentiated states [43] can be generated starting from an SMC state, as a precursor.

Lineage plasticity and transdifferentiation. Melanoma cells display an incredible plasticity. Different subsets of melanoma cells exhibiting a dedifferentiated state and stem-like properties, akin to their NCSC precursors, have been identified [45]. In particular, it has been demonstrated that these stem-like subpopulations display NCSC molecular features (i.e., KDM5B [45], CD133 [46,47], CD20 [46,48], NGFR [49,50] and AQP1), as well as biological properties such as high plasticity, migratory capacity, invasiveness and a general loss of pigmentation. Along with the lineage plasticity, transdifferentiation is another process described in melanoma cells, underlying their striking plasticity. Melanoma cells are able to transdifferentiate by exiting the melanocytic lineage to a different cell lineage like endothelial cells or CAFs [26,40], as illustrated in Figure 2.

Therefore, these subpopulations are able to adopt, transiently or permanently, different cellular states, each with implications on the proliferative abnormality, stromal reprogramming, angiogenesis, tumor sustaining inflammation and drug sensitivity, facilitating melanoma growth and progression. For instance, subpopulations of melanoma cells expressing high levels of EGFR and NGFR have been identified inside of tumors before therapy; it has been demonstrated that they are responsible for therapeutic relapse [49].

Moreover, melanoma cells are able to secrete growth factors and cytokines normally produced by stromal fibroblasts, macrophages, neutrophils and monocytes, promoting cell survival in an autocrine manner and influencing the tumor microenvironment in a paracrine loop. This mechanism, adopted by melanoma cells, is also known as stromal reprogramming.

### 1.3. Tumor Microenvironment

Tumor microenvironment (TME) plays an active role in melanomagenesis. For instance, overwhelming data on the critical role of the TME in melanoma progression have been reported, supporting the notion that melanoma cells alone are not able to cause disease, but rather need to corrupt and recruit neighboring healthy cell types to use as accessories to their evolution [51,52,53]. Therefore, the heterogeneous and plastic melanoma cells are only one part of a larger society comprising many other actors, defining the tumor microenvironment as a diversified compartment including neighboring stromal and non-stromal cellular components, the extracellular matrix (ECM) and soluble cues, as schematically illustrated in Figure 3.

Stroma includes the extracellular matrix (ECM, composed of glycoproteins, proteoglycans, glycosaminoglycans and other macromolecules [54]), growth factors and cytokines, the microvasculature, infiltrating inflammatory cells, and fibroblasts [55]. In melanoma, the stroma appears as desmoplastic (fibroblasts and fibrocytes with mainly fibrillar ECM components accumulation) or myxoid (atypical spindle cells with mainly proteoglycan accumulation [55]) and it is subjected to a constant remodeling by enzymes (collagenases and matrix metalloproteases, MMPs) and by fibroblasts. Interestingly, ECM stiffening, caused by increased collagen (mainly collagen I in the dermis) deposition and crosslinking, has been reported to disrupt the tissue structure, contributing not only to the malignant progression, facilitating tumor dissemination and metastasis, but also to the infiltration of immune cells in tumor sites [56,57]. The collagen fibers are arranged as vertical fibers in the papillary layer of the dermis, while those in the reticular layer are arranged parallel to the skin surface and are thicker.

Fibroblasts are involved not only in shaping ECM by producing its constituents as collagens and fibrous macromolecules and by degrading them, through releasing proteolytic enzymes, like MMPs; they are also a multifunctional cell type, playing a critical role in maintaining tissue homeostasis and in modulating the immune response. In fact, fibroblasts are involved in the leucocytes recruitment and in the regulation of inflammation, via the secretion of growth factors, cytokines and chemokines [58].

Stroma is activated in cancer, e.g., in wound healing, and fibroblasts inside resemble myofibroblasts observed during wound healing or fibrosis and are called cancer-associated fibroblasts (CAFs) [59,60].

Interestingly, distinct functional subsets of CAFs, exhibiting tumor-promoting or tumor-suppressing features, have been described, supporting that, as for cancer cells, CAF population is highly heterogeneous [61]. Furthermore, it has been demonstrated that, besides tissue resident fibroblasts, CAFs can also originate from mesenchymal stem cells (MSCs) or stellate cells [62], thus increasing their heterogeneity. MSCs are multipotent progenitor cells originating from the bone marrow. These cells migrate systemically through blood vessels and differentiate into osteoblasts, chondrocytes, or adipocytes. Moreover, they can also differentiate in vascular cells, contributing to angiogenesis in myofibroblasts and, more rarely, in cancer cells themselves. MSCs play a critical role in promoting tissue regeneration and, inside the TME, they modulate the immune response by releasing immunomodulatory cytokines.

The immune cells are another important component of TME. Various immune cell subsets have been identified infiltrating tumors. Among them, tumor-associated macrophages (TAMs) have been identified in all stages of tumor progression, showing antitumoral or pro-tumoral effects according to their inflammatory M1 or immuno-suppressive M2 phenotype, both depending on the microenvironmental stimuli [63,64].

Eosinophils, as TAMs, are also able to infiltrate tumors and influence tumor progression, inhibiting tumor growth by secreting IL-10 and IL-12, or promoting it by secreting growth factors such as epidermal growth factor (EGF) and transforming growth factor-b1 (TGF-b1) [65]. As tumors grow, myeloid-derived suppressor cells (MDSCs) [66], immunosuppressive precursors of macrophages and dendritic cells (DCs), stimulate the tumor vascularization and unsettle the major mechanisms of immunosurveillance, including tumoral antigen presentation, T cell activation and cytotoxicity.

T cells and natural killer (NK) cells represent the other major subset of tumor infiltrating immune cells. T lymphocytes include three major subtypes: (i) TH lymphocytes divided mainly in two lineages: pro-inflammatory TH1 and anti-inflammatory TH2; (ii) Regulatory T cells (Treg), primarily pro-tumorigenic via their immunosuppressive activity; and (iii) cytotoxic T cells (TC) that kill cancer cells through granzyme and perforin mediated apoptosis [67,68]. Moreover, a third lineage of effector TH lymphocytes, termed TH17 cells, characterized by their ability to secrete IL-17, have been identified and play a critical role in both anti-tumor immunity and tumorigenesis [69].

T cells are the major players in antitumor immune response. Thanks to the advanced technologies as well as multiplex immunohistochemistry methods [70], and mass cytometry (CyTOF) [71], it has been possible to get a comprehensive phenotyping picture of cells present in human tumor tissues. According to the cancer immunoediting, paradigm T cell infiltration edits the tumor during its progression and tumor evolution depends on the strength and quality of the local immune response at the metastatic site [72]. Intratumoral localization of T cells can be measured as ‘immunoscore’ value, and the high ‘immunoscore’ has been reported to be correlated with improved patient prognosis [71]. However, T cells can be also found outside the tumor [73,74], since it has been found that signaling pathways related to tumor cells (intrinsic pathways) or stromal components (extrinsic pathways) could induce T cells to become unable to enter in the tumor bed. This inability, also known as T cell exclusion process, has been indicated as a mechanism of resistance to cancer immunotherapy [75].

Recently, a computational framework has been created on the basis of Tumor Immune Dysfunction and Exclusion (TIDE) to identify factors related to the main mechanisms of tumor immune escape, which could represent a reliable surrogate biomarker to predict the immune checkpoint blockade (ICB) response [76]. Moreover, a signature associated with T cell exclusion and immune evasion has been defined by single-cell RNA sequencing (scRNAseq) of melanoma tumors; it has been demonstrated to be able to predict clinical responses to anti-PD-1 therapy [77].

Furthermore, another important cellular component of TME is represented by the endothelial cells, involved in angiogenesis [78,79] and vascularization, two important processes involved in cancer cell growth. These cells provide structural integrity to the newly formed vessels and, together with pericytes that ensure their coverage and maturity [80], promote the vascularization inside tumor bed. Endothelial cells not only create the roads for the metastatic dissemination via angiogenesis but they also contribute to chemotherapy resistance through an overexpression of drug efflux pumps thereby decreasing the tumor’s access to the drug [81].

Moreover, cancer-associated adipocytes (CAAs) support cancer growth mainly through secretion of adipokines like adipsin [82] or chemerin [83], as well as proinflammatory cytokines [84] and growth factors. CAAs constitute an important source of lipids for cancer cell membranes and organelles. They are involved in the metabolic reprogramming of cancer cells and in cancer cell invasion, as proteases suppliers [85]. It has been shown that, by producing tumor-promoting cytokines and factors, these cells are able to confer resistance to hormone therapies, chemotherapies, radiotherapies and targeted therapies in breast cancer [86], as well as to contribute to tumor progression of a variety of obesity-associated cancers [87] such as esophagus, gastric, liver, kidney, colorectal, pancreatic, breast, ovarian, prostate, and thyroid cancers. Further, adipocytes, from white adipose tissue, can be recruited to tumor sites where they can differentiate into pericytes and incorporate into vessel walls, thereby contributing to angiogenesis and to tumor proliferation [88].

Additionally, it has been observed that innervated tumors are very aggressive and highly proliferative, with an increased risk of recurrence and metastasis [89]. It is now evident that perineural invasion represents another route for dissemination [90]. Recently, it has been demonstrated that adrenergic nerves promote angiogenesis by activating the angiogenic switch in endothelial cells [91]. Moreover, several studies have described a process inside of a tumor termed axonogenesis, by which cancer cells stimulate the formation of new nerve endings within tumors, through the secretion of neurotrophic factors [92,93] or by releasing exosomes containing axonal guidance molecules [94]. In return, nerves provide the tumor with neurotransmitters that enhance its growth.

## 2. Melanoma Modeling

It is becoming more and more evident that in order to gain a better knowledge of melanoma evolution, but also to develop successful therapeutic strategies, it is necessary to faithfully recapitulate the in vivo human disease. Therefore, several efforts have been addressed in the development of ex vivo melanoma models able to capture the complex intra-tumor heterogeneity and plasticity in its environmental context. In this section, the ex vivo melanoma models used over time to investigate melanoma biology and its complexity are described and are schematically illustrated in Figure 4.

### 2.1. Ex Vivo Melanoma Models

#### 2.1.1. Two-Dimensional (2D) Melanoma Cell Culture

For several years, melanoma cells by themselves have been studied and largely characterized by using “traditional” melanoma cell lines established from patients and cultured on plastic (Figure 4a), with high levels of oxygen and nutrients.

Although this cell system does not provide information on cell-cell and cell–extracellular matrix interactions and on the tumor complexity, as well as on the melanoma behavior in vivo, this approach showed great utility in translational melanoma research. Comparative ‘omic’ studies aimed at characterizing melanoma cells established from melanoma patients at different clinical stages vs. melanocytes, and/or pre-malignant nevus cells able to identify biomarkers associated to the different cell lines examined [95,96,97].

Up to date, more than 2000 melanoma cell lines have been generated. Since these monocultures are free from other contaminating cells, the extensive genetic and genomic analysis that has been performed for the most of them, provided a comprehensive landscape of genes and pathways associated to melanoma progression and its drug resistance ability. In addition, the possibility to grow these cells in co-culture conditions, by using trans-well plates, allowed gaining a better understanding of the melanoma cell behavior upon a given insult.

Although there are several limits of the traditional 2D cell culture, such as the lack of heterogeneity, the different behavior of melanoma cells grown on plastic compared to the one observed in vivo, the high genetic variability with consequent lack of reproducibility of the experimental results depending on the genetic drift, occurring in long-term passaged cell lines, this approach remains very useful for the initial high-throughput screening to identify potential hits worth of further examination.

#### 2.1.2. Three-Dimensional (3D) Melanoma Cell Culture

Multicellular Spheroids. Multicellular Spheroids (MCSs) consist of 3D cellular aggregates of homogeneous or heterogeneous cell populations derived from tissue fragments mechanically and/or enzymatically partially digested, as illustrated in Figure 4b. MCSs are obtained in the absence of a scaffolding material, as cultured cells produce their own ECM and can be used to generate either homogeneous tumor models by starting from solely cancer cells cultures, or more sophisticated heterotypic spheroids by starting from cancer cells cultures with components of the TME like fibroblasts, endothelial cells [98] or immune cells. Different techniques have been developed to generate these models in laboratory, such as the forced floating methods in non-adherent plates, the hanging drop method, the use of scaffolds and matrices, or even more sophisticated methods using microfluidic systems [99].

Current three-dimensional melanoma models are composed of melanoma cells only (melanoma spheroids) [100] or they are more sophisticated, including multiple cell types to reproduce human skin equivalents with skin-like organization. An intermediate spheroid-based model has also been developed, consisting of tri-cultures of human fibroblasts, keratinocytes, and melanoma cells. These systems have the advantage of being reliably reproduced without the need of special equipment, since they are made of cell lines and therefore can be generated by using standard culture procedures. Moreover, this kind of model can be helpful to investigate the different aspects of skin and early melanoma formation as to test drug compounds [101]. Further, melanoma multicellular spheroids model composed of melanoma cells, fibroblasts, and macrophages have been generated by liquid-overlay technique using agarose gel. These models have been helpful in the examination of stromal cell influence on their size, growth, viability and morphology, compared to the melanoma monocellular spheroids [102].

Recently, several efforts are addressed to developing biomimetic hydrogel scaffolds that can then be used to encapsulate the MCSs in order to provide a more complex tumor model, where the biophysical and biochemical cues are enclosed, simulating the behavior of the ECM, that it is known to play an important role in the modulation of cancer cell behavior [103].

Although these models recapitulate the TME heterogeneity [104], oxygen gradients [105], and immune infiltration [106], they are limited by the lack of control over the 3D culture environment, being that they are generated by the self-assembling of cells.

3D skin reconstructs. This 3D model captures the melanoma heterogeneity and the complex intra-cellular interactions similar to the one occurring in in vivo human disease. It includes an “epidermis” containing stratified, differentiated keratinocytes, a functional basement membrane, and a “dermis” with fibroblasts embedded in collagen I, the most prevalent extracellular matrix (ECM) present in the human skin [107] (Figure 4c). However, in order to generate 3D skin reconstructs in the laboratory, the ability to obtain melanoma cells, keratinocytes, fibroblasts and melanocytes in viable culture is critical. Fibroblasts and melanocytes can also be derived from human skin, but can either come from embryonic stem cells (ESCs) [108] or induced pluripotent stem (iPS) cells [109]. The 3D skin reconstruct models are helpful tools for invasion and metastasis studies as well as for analysis of drug effects on melanoma cells [110].

Organotypic Melanoma Models Like Organoids. 3D organotypic melanoma models have been developed [111] to reproduce ex vivo the complexity of melanoma (Figure 4d). They are considered the more physiological 3D culture models. Similar to tissue like organoids [112,113,114], they represent an innovative approach for melanoma modeling studies and anticancer drug testing. Several efforts are ongoing in order to develop novel synthetic analogous ECM, controllable for allowing a fine tuning of matrix constituents [115]. These approaches permit one to mimic the organ topography, the cancer cells’ mechanical forces, the stiffness, functionality, and complexity of matrix much better than 2D or even 3D culture systems [116].

Skin-on-chip. Since organotypic melanoma models lack parameters such as fluid shear stress and hydrostatic pressure, which are able to greatly influence cell behavior in the physiological conditions, several efforts have been addressed into the development of microfluidic systems [117]. These cell culture systems, also known as organ-on-a-chip, are made of hollow microchannels populated by living cells and continuously perfused [118]. To date, skin-on-chip [119] have been successfully modeled in microfluidic devices (Figure 4e), as well as lung alveoli [120], human kidney tubules [121], and liver [122]. These systems show the big advantage to reproduce a spatio-temporally controlled microenvironment, where all the molecular, biophysical and cellular components can be tuned according to the physiologically relevant parameters Furthermore, they represent a feasible tool for drug efficiency and toxicity assessment.

Ex vivo tissue slices [123]. They represent a further tool, by which the tissue 3D architecture and pathway activity is preserved although for short time [124] (Figure 4f). This tool has been revealed to feasibly track T cells and identifying the extracellular matrix as the major stromal component influencing T cell migration in fresh human tumor tissues [125]. Furthermore, the analysis of ex vivo tissue slices by dynamic imaging microscopy allowed us to highlight the mechanism underlying T cell exclusion by examining the interaction between endogenous CD8 T cells and tumor-associated macrophages (TAMs) inside the tumor stroma. These studies translated in a murine model showed that TAMs depletion enhanced the efficacy of anti–PD-1 immunotherapy [126]. Thus, this tool may be used in studies of screening for novel immunotherapy agents and in the T cells monitoring inside of the tumor.

## 3. Melanoma Immunotherapy and Precision Medicine: Where We Are Today

Although immunotherapy has drastically changed the clinical management of metastatic melanoma, most patients treated with checkpoint inhibitors (CPIs) do not respond. About 79% of metastatic melanoma patients treated with ipilimumab die within 5 years, while patients treated with the ipilimumab plus nivolumab combined therapy, show a median progression-free survival of 11.5 months compared to 2.9 months observed in patients treated only with ipilimumab [127,128]. Furthermore, it is not easy to predict which metastatic melanoma patients will be able to respond to with immunotherapy because of the lack of deep understanding of the cellular and molecular mechanisms that lead to PD-1 blockade resistance. Being that this is not the topic of this review, the reader is referred to a recent review on a comprehensive description of the primary and acquired resistance to immune CPIs in metastatic melanoma [129].

Therefore, to move forward with more efficacious personalized treatment and precision medicine, not only predictive markers of response to therapy are under investigation but more pre-clinical models are under development.

Recently, several efforts have been addressed to identify predictive biomarkers of clinical response. Interestingly, gene sequencing studies have discovered markers for monitoring anti-tumor response and therapeutic outcomes after PD-1 blockade failure, like TMB, neoantigen load (NL) or PDL1 expression degree, often associated with an increased response to immunotherapy [130,131]. Furthermore, a new therapy recently evaluated for melanoma treatment is the oncolytic virus anti-cancer therapy. This therapeutic strategy is based on the ability of oncolytic virus to indirectly lysate tumor cells, leading to the release of soluble antigens and interferons, driving the antitumor immunity. In particular, the attenuated herpes simplex virus-based oncolytic virus talimogene laherparepvec (T-VEC) was FDA approved in 2015, and it is currently used as a local treatment of patients carrying an unresectable advanced stage melanoma [132]. Interestingly, being that oncolytic viruses are able to induce immunogenic tumor cell death, their use in combination with ICIs may represent an interesting and promising strategy for melanoma patients treatment [133].

Similarly, it has been observed that the activation of NF-kB (nuclear factor kappa-light-chain-enhancer of activated B cell) signaling represents a novel potential marker of response to immunotherapy in metastatic melanoma [134]. In particular, it has been observed that a higher mutational load of NFKBIE (NF-kB negative regulator), in codons G34 and G41, only in patients, who were more responsive to anti-PD1 therapy. NFKBIE loss of function culminated in the activation of the NF-kB pathway, which, therefore, can be considered a possible predictive factor of treatment response [135]. Moreover, alterations to DNA damage repair (DDR) pathways have been found associated with a better response to immune checkpoint inhibitors (ICIs).

Furthermore, melanoma heterogeneity is currently treated as another critical parameter of response to immunotherapy. Several studies have shown that patients with less heterogeneous melanoma responded better to the blocking action of anti-CTLA-4 and anti-PD1, supporting the concept that the high heterogeneity implies a major presence of tumor subclones, able to bypass the immune system and thus drug resistance [136].

In addition to the various factors already described, a study conducted on 144 patients with metastatic melanoma included purity and ploidy of the tumor as predictive markers of response to PD1 inhibitors. Specifically, higher tumor purity was associated with tumor progression, while the ploidy was lower in non-responding patients.

The clinical management following immunotherapy failure remains challenging, and a precision medicine (based on specific markers and mutations) is desperately needed in order to create a more personalized treatment [137]. It will help to choose the optimal therapeutic strategies and to predict the consequences of a medical treatment. For instance, the genomic profiling helps to profile the patients and to separate them with the same diagnosis into different groups based on the knowledge of the molecular and cellular mechanisms of the disease [138].

Moreover, moving forward, a precision medicine requires the development of more complex cellular models able to recapitulate more closely the melanoma in human patients [139]; nowadays we are witnessing advances in designing tumor micro-tissues simulating in vivo situations.

## 4. Melanoma Immunotherapy and Precision Medicine: Where We Are Going in the Tissue Micro-Engineering Era

Advances in the development of melanoma models capable of maintaining the in vivo physiological pressures, where melanoma cells behave as they would in human patients, are offering to the researcher new tools and approaches to better investigate the melanoma biology. Several studies have reported the critical role of TME in modulating T-cell function, particularly in response to PD-1 blockade during melanoma treatment [140] highlighting the necessity of more sophisticated experimental tumor models incorporating key features of the native immune TME, that can be analyzed in real time in order to drive translational research efforts in the clinic.

Recently, human organotypic skin melanoma cultures (OMC) have been developed, by co-culturing decellularized dermis with keratinocytes, fibroblasts and immune cells in the presence of melanoma cells [141]. Interestingly, these human OMCs have been demonstrated to be able to mimic the natural primary human melanoma lesions as well as to be feasible for studying the TME-imprinting mechanisms responsible of melanoma progression. In particular, by using these OMCs, it was demonstrated that the immune cells cDC2s (type 2 conventional dendritic cells) in the TME were melanoma-driven converted into CD14b + DCs.

cDC2s are phenotypically defined as CD1c+CD14− and are able to stimulate cytotoxic T-cell responses [142]. Interestingly, it has been observed that these immune responsive cells were melanoma-induced and converted in CD14+ DCs. These cells are characterized by the expression of genes, such as SSP1, PTGS2 and IL-6, which have been previously associated with immunosuppressive myeloid cells [143,144], like monocytes and macrophages, having poor T-cell stimulatory ability. Furthermore, the reprogramming of mature cDC2s into CD14+ DCs regulatory macrophage-like cells suggested in this study has been previously proposed only in murine models [145,146], since the blood cDC2s in human healthy individuals exhibit a low heterogeneity as revealed by single-cell-RNA sequence, and therefore it may not explain the cDC2s phenotypic plasticity observed in these OMCs. Importantly, this study introduced a new tool to use in order to analyze the DCs dynamic interactions with tumor cells inside of the reconstructed TME. This tool could also be used in the future to better define the mechanisms modulating the fate of individual DC subsets within the TME, as well as the migratory nature of DCs towards tumor cells as a stochastic or a chemotactic gradient-driven process induced by the melanoma cells.

Moreover, another study revealed how the organotypic tumor models are powerful tools to evaluate immunotherapy efficacy. In this case, organotypic tumor spheroids grown in collagen hydrogels in a 3-D microfluidic culture system [147] were developed.

In particular, patient- and murine-derived tumor spheroids (MDOTS/PDOTS), retaining tumor-infiltrating lymphoid and myeloid subpopulations, were generated and then used in short-term ex vivo culture to analyze their response to PD-1 blockade treatment, by profiling the secreted cytokines upon ICB treatment [147]. Interestingly, although preliminary data, given the relatively small cohort of samples, a clear relationship between CCL19/CXCL13 cytokines production and immune infiltration was observed. CCL19 has been already reported to be produced from cancer-associated fibroblasts (CAF), while CXCL13 from CD8+ exhausted T cells [40] in melanoma specimens by using single-cell RNA sequencing (RNA-seq). Importantly, this study suggests that both cytokines in PDOTS models may recruit immune- suppressive cells and act as intrinsic resistance mediators to PD-1 blockade.

Actually, CCL19 and CXCL13 cytokines have already been reported to coordinate both humoral and cell-mediated adaptive antitumor immune responses by facilitating the recruitment of naïve T cells and dendritic (CCR7+) and specific B- and T-cell subsets (CXCR5+) to the sites of chronic inflammation [40,148,149]. Instead, in this study, these cytokines have been identified as shared acute cytokines to PD1-blockade response, suggesting that future studies need to be performed to highlight the differences between the early and late events of immune response.

Moreover, thanks to the continuous advances in micro-tissue engineering, Votanopoulos et al. were able to generate 3D mixed tumor/node organoids from melanoma patients. The Authors demonstrated that these human experimental models were able to recapitulate the interaction between tumor, host and immune system, representing a feasible platform for personalized immunotherapy screening [150]. In particular, they generated patient-specific immune-enhanced organoids (iPTOs), starting from ten matched melanomas (stage III and IV) and lymph node biospecimens, obtained from the same patient. Further, where it was not possible to obtain lymph nodes from patients, mixed tumor/peripheral T cell organoids were generated starting from the peripheral blood T cell component of the same patient, where tumor was resected.

Additionally, the Authors demonstrated that, when peripheral T cells were circulated through iPTOs and subsequently transferred to naïve PTOs from the same patient, they become able to kill cancer cells, thus suggesting a possible role of iPTOs in generating adaptive immunity.

Although these are preliminary data, given the small numbers of iPTOs examined, they are very promising. It was found a correlation of 85% (6 on 7 patients) between response to immunotherapy observed in the iPTOs and the clinical response of the corresponding patient. Six patients showed melanoma progression while on treatment as their corresponding iPTOs.

All together, these studies demonstrate the enormous potential of ex vivo testing in patient-derived tumor spheroids to identify effective therapeutic combinations to overcome intrinsic resistance to PD-1 blockade. Thus, future adaptations of these models may provide a useful functional approach to drive clinical-translational efforts leading to personalized immunotherapy, lowering its cost, and increasing its effectiveness.

## 5. Conclusions

It is clear that melanoma is a complex disease, characterized by high heterogeneity and plasticity. Melanoma cells are only a part of a large ecosystem where tumor microenvironment plays an active part on their evolution and on their ability to escape to drug treatment. Therefore, recently several efforts have been addressed to the development of ex-vivo models able to recapitulate the live conditions of melanoma cells in human patients, in order to gain a better understanding of the mechanisms underlying melanoma biology and therapeutic resistance.

Complex 3D-models in microfluidic systems able to mimic the melanoma immune microenvironment have been developed and used as a novel approach by the researcher. Although more complex patient-derived xenografts (PDX) models have been generated [151] to keep the complexity of human tumors, these models are limited by the fact that over time their stromal compartment turns in the murine host one, making more complex the immunotherapy studies [152]. For this reason, in order to investigate on the immune and stromal compartments of the tumor microenvironment, the ex-vivo micro-tissue models are preferred to the PDX ones. Moreover, ex-vivo micro-tissue models have the big advantage to be more efficient and lower in cost than the maintenance of large-scale animal colonies. However, they present limitations in immune cell viability over long-term culture. Thus, more efforts in this field are needed to identify strategies to optimally maintain the various immune populations observed in the tumor microenvironment [153,154].

Combining ex-vivo micro-tissue models with other techniques, such as single-cell sequencing or advanced microscopy methods, will allow the researcher to highlight our knowledge in immune-tumor cell interactions and immunotherapy and presents the huge potential to pave the way for translational personalized medicine. Therefore, it will be critical to design ex-vivo micro-tissue models compatible with high-throughput molecular analysis such as gene sequencing or mass spectrometry. High-throughput and single-cell gene expression profiling will provide a better understanding of the evolution of tumor cell heterogeneity, as well as of immune landscape dynamicity, when patient-derived organotypic models are cultured within a microfluidic TME.

Finally, in order to use these ex-vivo models in a microfluidic TME as a surrogate for in vivo pre-clinical testing, more studies need to be addressed toward the validation and improved consistency of results. Therefore, it will be critical to repeat studies such as the one from Votanopoulos et al., where the correlation between response to PD-1 blockade in patient-specific immune-enhanced organoids (iPTOs), and in vivo response of the same patient and drug was examined [150]. Patient-derived organotypic models will also likely become a useful tool, particularly for identifying efficacious therapeutic regimes [155]. The establishment of biobanks of patient-derived organoids combined with matched peripheral blood samples for the isolation of circulating immune cells will further support translational research [156,157,158]. Moreover, these ex-vivo micro-tissue models will be another tool for the researcher, helpful not only to investigate the influence of the microenvironment in tumor progression, but at the same time to allow the researcher to combine, include, or exclude particular TME cell types as well as more straightforward hypothesis testing than using animal models [159].

## Data Availability

Not applicable.

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
