# Peer review of "Melanoma Immunotherapy and Precision Medicine in the Era of Tumor Micro-Tissue Engineering: Where Are We Now and Where Are We Going?"

_cancers, 2021, doi:10.3390/cancers13225788_

Round 1

Reviewer 1 Report

Despite the manuscript presenting an interesting overview of melanoma pathogenesis and therapy, some of the material is lacking specific information. Despite in most of the cases, this info is not novel and well-established, it will be much better to make it available for a broad audience.

For example, the authors should include the info about the drugs (e.g. RTK inhibitors and immune checkpoint inhibitors, as well) when they describe these therapeutic approaches that drastically improved the outcomes for patients with melanoma, especially in metastatic settings (lines 49-52). Despite the authors mentioning that this is not the topic of this review (line 479), the title of the manuscript highlights the immunotherapeutic approaches for patients with melanoma and I believe that providing the Table illustrating the current therapeutic regimens used for melanoma treatment. 

Similarly, the authors mention about driver mutations play an important role in melanoma pathogenesis (lines 80-82). Again, despite they are well-known, but for the review, it will be much better to have some specific info in the text in a separate table. 

Author Response

Dear Reviewer 1:

I appreciated your constructive criticisms and I have addressed each of your concerns as outlined below.

Please you will find all the revisions highlighted in red in the manuscript.

My responses to the your comments (normal font) are in italic font.

REVIEWER 1

Comments and Suggestions for Authors

1. For example, the authors should include the info about the drugs (e.g. RTK inhibitors and immune checkpoint inhibitors, as well) when they describe these therapeutic approaches that drastically improved the outcomes for patients with melanoma, especially in metastatic settings (lines 49-52). Despite the authors mentioning that this is not the topic of this review (line 479), the title of the manuscript highlights the immunotherapeutic approaches for patients with melanoma and I believe that providing the Table illustrating the current therapeutic regimens used for melanoma treatment. 

Information about target-based drugs and immuno-therapeutics have been included in the text (lines 51-61, 87-90, 95-102), in Table 1 (page 3), in Scheme 1 and Table 2 (page 4).

2. Similarly, the authors mention about driver mutations play an important role in melanoma pathogenesis (lines 80-82). Again, despite they are well-known, but for the review, it will be much better to have some specific info in the text in a separate table. 

Specific info on driver mutations have been added in the text (lines 51-61) and in table 1 (page 3).

Reviewer 2 Report

The review authored by Varrone et al is about an interesting and growing epidemic which is represented by melanoma

1) A brief description of melanoma mutations (BRAF, NRAS) and available treatment strategies is missing

2) T-VEC is the first oncolytic adenovirus approved both in Europe and US for the treatment of melanoma. This must be reported in the immunotherapy´s chapter

3) Also the usage of oncolytic viruses in combination with ICIs is an interesting and promising approach for patients

Minor

resolution of figure 3 has to be improved

Author Response

Dear Reviewer 2:

I appreciated your constructive criticisms and I have addressed each of your concerns as outlined below.

Please you will find all the revisions highlighted in red in the manuscript.

My responses to the your comments (normal font) are in italic font.

REVIEWER 2

Comments and Suggestions for Authors

  • A brief description of melanoma mutations (BRAF, NRAS) and available treatment strategies is missing.

Both information has been included in the text (lines 51-61) and in table 1 (page 3).

  • T-VEC is the first oncolytic adenovirus approved both in Europe and US for the treatment of melanoma. This must be reported in the immunotherapy´s chapter

This info has been reported in the immunotherapy´s chapter (lines 575-583) as well as in Table 2 (page 4).

  • Also the usage of oncolytic viruses in combination with ICIs is an interesting and promising approach for patients

It has been included in the immunotherapy´s chapter (lines 581-583).

Minor

resolution of figure 3 has to be improved

The resolution of Figure 3 has been improved (page 9).

Round 2

Reviewer 2 Report

Authors replied to my comments. I suggest the MS for publication